# Cell Surface Hsp90- and αMβ2 Integrin-Mediated Uptake of Bacterial Flagellins to Activate Inflammasomes by Human Macrophages

**DOI:** 10.3390/cells11182878

**Published:** 2022-09-15

**Authors:** Thi Xoan Hoang, Jae Young Kim

**Affiliations:** Department of Life Science, Gachon University, Seongnam 13120, Korea

**Keywords:** all-trans retinoic acid, flagellin, heat shock protein90, αMβ2 integrin, inflammasome

## Abstract

All-trans retinoic acid (ATRA) is an active metabolite of vitamin A, which plays an important role in the immune function. Here, we demonstrated that ATRA induces the heat shock protein (Hsp) 90 complex on the surface of THP-1 macrophages, which facilitates the internalization of exogenous bacterial flagellins to activate the inflammasome response. Mass spectrometric protein identification and co-immunoprecipitation revealed that the Hsp90 homodimer interacts with both Hsp70 and αMβ2 integrin. ATRA-induced complex formation was dependent on the retinoic acid receptor (RAR)/retinoid X receptor (RXR) pathway and intracellular calcium level and was essential for triggering the internalization of bacterial flagellin, which was clathrin dependent. Notably, in this process, αMβ2 integrin was found to act as a carrier to deliver flagellin to the cytosol to activate the inflammasome, leading to caspase-1 activity and secretion of interleukin (IL)-1β. Our study provides new insights into the underlying molecular mechanism by which exogenous bacterial flagellins are delivered into host cells without a bacterial transport system, as well as the mechanism by which vitamin A contributes to enhancing the human macrophage function to detect and respond to bacterial infection.

## 1. Introduction

Heat shock protein (Hsp) 90 is an ATP-dependent molecular chaperone that plays a role in maintaining the conformational integrity of various proteins [1]. Hsp90 consists of three distinct domains: an N-terminal ATP-binding domain that plays a crucial role in protein function, the middle domain with a client-protein-binding site and a C-terminal dimerization domain. In addition to the cytosolic Hsp90, previous studies have uncovered the cell surface and secreted Hsp90 in the surrounding environment where it facilitates the uptake of antigens for presentation [2,3,4]. By interacting with different cell surface receptors, such as scavenger receptors expressed by endothelial cell-I (SREC-I) and lectin-like oxidized low-density lipoprotein receptor-1 (LOX-1), eHsp90α facilitates the uptake and processing of antigens by antigen presenting cells (APC) for presentation to T cells, either through the major histocompatibility complex (MHC) class II [4] or class I pathways [2]. In these studies, nonpathogenic antigens, such as tumor antigens or ovalbumin, were used. Intracellular Hsp90 mediates the translocation of bacterial toxins from the endosomes to the cytosol. In this process, two different immunophilins, cyclophilin and FK506-binding protein, appear to act as accessory proteins to facilitate the transmembrane transport of Clostridium ADP-ribosylating toxins [5]. In other studies, thioredoxin reductase [6] and cyclophilins [7] also facilitated Hsp90-mediated transport of the enzyme subunit of diphtheria toxin across endosomal membranes. In addition to Hsp90, Hsp70 was found to facilitate the transport of Clostridium ADP-ribosylating toxins across the endosomal membrane [8].

All-trans retinoic acid (ATRA) is an active metabolite of vitamin A, which plays an important role in various cell fate determining processes, such as cell proliferation, differentiation and apoptosis [9]. In addition, ATRA plays a critical role in homeostatic control of the immune system. ATRA promotes the generation of mucosal dendritic cells (DCs), regulates the differentiation and function of CD4+ T cells and induces gut-homing T and B cells [10]. In addition to these functions, our recent studies have demonstrated the immune-potentiating effect of ATRA on human monocytes/macrophages [11,12]. Owing to its crucial immune regulatory functions, vitamin A is essential for resistance to infections; therefore, vitamin-A-deficient individuals are more susceptible to infection [13]. Despite its profound functions in the immune system via the regulation of immune cells fate and function, the detailed molecular mechanism by which ATRA promotes immune cell defense against infectious antigens still remains to be elucidated.

Flagellin is a structural component of the bacterial flagellum and is considered to be a pathogen-associated molecular pattern (PAMP). Flagellin is recognized by the cell surface Toll-like receptor (TLR) 5 of immune cells, such as macrophages [14]. Upon stimulation of TLR5 by flagellin, a cytoplasmic Toll/IL-1 receptor domain recruits intracellular signaling molecules to activate the transcription of pro-inflammatory genes [15]. It can also be detected by the cytoplasmic receptors, NLR family apoptosis inhibitory protein (NAIP) 5 and NAIP6 [16], if it is delivered to the cytoplasm. Some Gram-negative bacteria, including Salmonella typhimurium, can deliver exogenous flagellin into the cytoplasm through a type III secretory system (T3SS) [17]. The flagellin recognized by the NAIP5/6 complexes subsequently interacts with the nucleotide-binding oligomerization domain protein (NOD)-like receptor NLR family CARD domain containing 4 (NLRC4), resulting in caspase-1 recruitment to the CARD domain to form an inflammasome complex, and activates caspase-1, which cleaves pro-interleukin (IL)-1β into active IL-1β [18]. Thus, to examine the response of NAIP/NLRC4 inflammasomes to exogenous purified flagellin in the absence of T3SS, other delivery system(s), such as cationic liposomes, are necessary [19]. However, in other in vitro studies, including ours, exogenous flagellin was found to exert such inflammasome response even in the absence of any delivery system [11,20], implying the existence of an active delivery mode of host cells. 

Here, we demonstrate that ATRA induces the cell surface Hsp90α complex, which facilitates αMβ2 integrin-mediated uptake of bacterial flagellins to activate the inflammasome response in the human macrophage cell line, THP-1. 

## 2. Materials and Methods

### 2.1. Chemicals 

The following chemicals were used: ATRA, RAR α agonist (BMS753), RXR α agonist (LG100268), RARα antagonist (BMS195614), GA, EGS, lipid raft inhibitor filipin III (F-4767), caveolin inhibitor methyl-beta-cyclodextrin (C-4555) from Sigma-Aldrich (St. Louis, MO, USA); bis (sulfo-succinimidyl) suberate (BS3) from Thermo Fisher Scientific (Rockford, IL, USA); RXRα antagonist (UVI 3003) from Santa Cruz Biotechnology (Dallas, TX, USA); flagellin from S. typhimurium, GeB from InvivoGen (San Diego, CA, USA); 17-AAG and radicicol from Cayman Chemical (Ann Arbor, MI, USA); dynamin inhibitor dynasore (ab120192), Dyngo^®^ 4a (ab120689) from Abcam (Cambridge, UK). 

### 2.2. Cell Culture 

The human monocytic cell line, THP-1 (Korean Cell Line Bank, Seoul, Korea), was grown in RPMI-1640 media (Welgene Inc., Gyongsan, Korea) supplemented with 10 mM HEPES buffer ((Invitrogen, Gibco BRL, Gaithersburg, MD, USA), β-mercaptoethanol (Sigma-Aldrich, St. Louis, MO, USA) supplemented with 10% heat-inactivated fetal bovine serum (FBS, Invitrogen, Gibco BRL, Gaithersburg, MD, USA) and 1% antibiotic-antimycotic (Invitrogen, Gibco BRL, Gaithersburg, MD, USA). The cells were maintained at 37 °C in a 5% CO_2_ humidified incubator (NuAire, Plymouth, MA, USA). 

### 2.3. RNA Preparation and Real-Time Quantitative Polymerase Chain Reaction (qPCR)

Total RNA was extracted using the Qiagen RNAeasy Mini Kit (Qiagen, Hilden, Germany) and reverse transcribed into cDNA with Hyperscript RT master mix (GeneAll, Seoul, Korea) using an Oligo (dT) primer (Invitrogen, Gibco BRL, MD, USA) according to the manufacturer’s instructions. Then, qPCR was performed on a Rotor-gene system (Qiagen) using the Platinum SYBR Green qPCR SuperMix-UDG (Invitrogen, Corp., Carlsbad, CA, USA). The following primer sets were used: Hsp90-α 5′-CGTCTTCGGAAACATGGCTT-3′, 5′-CGGTTTGACACAACCACCTT-3′, IL-1β 5′-GGGATAACGAGGCTTATGTGC-3′, 5′-AGGTGGAGAGCTTTCAGTTCA-3′, caspase-1 5′-CGATTTTCATTTGAGCAGCCA-3′, 5′-ATCTCTTCACTTCCTGCCCAC-3′ and β-actin 5′-CACCATTGGCAATGAGCGGTTC-3′, 5′-AGGTCTTTGCGGATGTCCACGT-3′. Sample normalization was performed using the human β-actin gene as an endogenous control. For each sample, the relative abundance of target mRNA was calculated from the C△t values of the target and endogenous β-actin reference genes using the 2−△△cycle threshold (Ct) method. 

### 2.4. Flow Cytometry

To determine the cell surface expression, cells were incubated with primary antibodies, followed by incubation with phycoerythrin (PE)-conjugated goat anti-mouse IgG antibody (ab97024, Abcam, Cambridge, UK) or PE-conjugated donkey anti-goat IgG antibody (ab7004, Abcam), and analyzed on a Cytomics FC500 MLP (Beckman Coulter, Fullerton, CA, USA). The following antibodies were used: Hsp90 (sc-69703), αM integrin (ICRF44) from IQ Products (Groningen, The Netherlands); β2 integrin (sc-8420) from Santa Cruz; Hsp70 (ab2787) from Abcam (Cambridge, MA, USA); and flagellin (mabg-flic) from Invivogen (San Diego, CA, USA). To detect the intracellular expression, cells were suspended in the permeabilization buffer (eBioscience, San Diego, CA, USA), centrifuged and stained with primary antibodies diluted in the permeabilization buffer. After incubation with PE-conjugated secondary antibodies, the fluorescent signal was measured by Cytomics FC500 MLP. 

To examine the intracellular Ca^2+^ levels, cells were seeded onto 6-well culture plates at a density of 1 × 10^5^ cells/well. After treatment, the cells were collected and washed with calcium-free DPBS (Sigma-Aldrich). The cells were then incubated with 50 µL Flou-4-NW-dye for 30 min at 37 °C in the dark, followed by incubation at 25 °C for 30 min. The fluorescence was measured by Cytomics FC500 MLP. 

### 2.5. Cell Surface Biotinylation

The Pierce^®^ Cell Surface Protein Isolation Kit was used for the isolation and collection of surface proteins. Briefly, following specific treatments, THP-1 cells were washed with ice-cold PBS and then incubated with 0.25 mg/mL Sulfo-NHS-SS-Biotin in 48 mL ice-cold PBS on a rocking platform for 30 min at 4 °C. The clarified protein eluate was used for the purification of biotinylated proteins on pre-cleared NeutrAvidin agarose. The protein eluate was added to the slurry and incubated for 2 h at 4 °C in a closed column using an end-over-end tumbler to mix vigorously and allow the biotinylated proteins to bind to the NeutrAvidin agarose slurry. The unbound proteins, representing the intracellular fraction, were collected by centrifugation and stored at −20 °C to serve as an internal control for the surface protein isolation process. Finally, the captured surface proteins were eluted from the biotin-NeutrAvidin agarose by incubation with SDS-PAGE sample buffer containing 50 mM dithiothreitol (DTT, Thermo Fisher Scientific, Rockford, IL, USA). The eluted proteins, representing the cell surface proteins, were collected by column centrifugation.

### 2.6. Western Blotting Analysis 

Cells were lysed with the Triton X-100 lysis buffer (Sigma-Aldrich). The supernatants from whole-cell lysates were collected after centrifugation. Denatured proteins were subjected to 8% SDS-PAGE and blotted onto a polyvinylidene difluoride membrane. The membranes were blocked with 5% bovine serum albumin (BSA, Sigma-Aldrich), followed by probing with the following primary antibodies: Hsp90 (sc-69703), αM integrin (ICRF44) from IQ Products; β2 integrin (sc-8420), caspase-1 (sc-56036) and β-actin (sc-47778) from Santa Cruz; Hsp70 (ab2787) from Abcam. The membranes were then incubated with horseradish peroxidase-conjugated anti-mouse IgG antibody (sc-2055), or anti-goat IgG antibody (sc-2020) from Santa Cruz, and visualized with the ECL solution using the ChemiDoc MP system (Bio-Rad) according to the manufacturer’s protocol. β-actin was used as a positive control for sample normalization.

### 2.7. Cross-Linking Reaction 

Amine-specific homobifunctional cross-linkers including membrane-impermeable BS3 and membrane-permeable EGS were used to cross-link the lysine residues in amine groups of protein molecules. Cross-linking reactions were performed according to the manufacturer’s instructions. Briefly, cells were incubated with 2.5 mM cross-linkers for 30 min at 4 °C. The reactions were quenched by the addition of 1 M Tris to a final concentration of 10–20 mM for 15 min at RT. Cells were then lysed and subjected to Western blotting analysis.

### 2.8. Co-Immunoprecipitation

Co-IP was carried out using the Pierce Co-IP kit according to the manufacturer’s instructions. Briefly, 1 mg of cell lysate was pre-cleared by incubation with Protein G Sepharose 4 Fast Flow media at 4 °C and with rotation for 1 h. Subsequently, pre-cleared lysates were incubated with the primary antibody or IgG as a negative control with gentle end-over-end mixing at 4 °C overnight. The mixture was then collected by centrifugation at 1000× *g* for 1 min. The flow-through was saved for unbound protein verification. The collected beads were washed three times with the immunoprecipitation buffer. Finally, the bound proteins were eluted using an immunoprecipitation elution buffer. The collected eluates were used for Coomassie staining and subsequent protein excising or Western blotting analysis. 

### 2.9. In-Gel Digestion with Trypsin and Extraction of Peptides

Protein bands from the SDS-PAGE gels were excised and in-gel digested with trypsin (Promega, Madison, WI, USA). Briefly, protein bands were excised from the stained gels and cut into pieces. The gel pieces were washed for 1 h at RT in 25 mM ammonium bicarbonate buffer (pH 7.8) containing 50% (*v*/*v*) acetonitrile (ACN, Sigma-Aldrich). Following the dehydration of gel pieces in a centrifugal vacuum concentrator (Biotron, Korea) for 10 min, the gel pieces were rehydrated in sequencing-grade trypsin solution (Promega). After incubation in 25 mM ammonium bicarbonate buffer at 37 °C overnight, the tryptic peptides were extracted with 1% formic acid (FA, Sigma-Aldrich) containing 50% (*v*/*v*) ACN for 20 min with mild sonication. The extracted solution was concentrated using a centrifugal vacuum concentrator. Prior to mass spectrometric analysis, the peptide solution was subjected to a desalting process using a reversed-phase column. 

### 2.10. Identification of Proteins by Liquid Chromatography–Tandem Mass Spectrometry (LC-MS/MS)

LC-MS/MS analysis was performed using a nano ACQUITY UPLC and LTQ-orbitrap-mass spectrometer (Thermo Electron). Mobile phase A for LC separation was 0.1% FA in deionized water, and mobile phase B was 0.1% FA in ACN. The chromatography gradient was set up to give a linear increase from 10% B to 40% B for 21 min, from 40% B to 95% B for 7 min and from 90% B to 10% B for 10 min. The flow rate was 0.5 µL/min. For MS/MS, mass spectra were acquired using data-dependent acquisition with a full mass scan (300–2000 m/z) followed by MS/MS scans. Each MS/MS scan acquired was the average of one microscan on the LTQ. The temperature of the ion transfer tube was controlled at 160 °C, and the spray was 1.5–2.0 kV. The normalized collision energy was set to 35% for the MS/MS analysis. The individual spectra from MS/MS were processed using the SEQUEST software (Thermo Quest, San Jose, CA, USA), and the generated peak lists were used to query the in-house database using the MASCOT program (Matrix Science). MS/MS ion mass tolerance was 0.8 Da, the allowance of missed cleavage was 2, and charge states (+2, +3) were taken into account for data analysis. Significant hits were defined by the MASCOT probability analysis. The mass spectrometry data were deposited at the ProteomeXchange Consortium (https://doi.org/10.25345/C54W0Q accessed on 24 January 2022). 

### 2.11. Immunofluorescence

Cells were fixed with 4% formaldehyde in PBS, followed by permeabilization with 0.1% Triton X-100 for intracellular protein staining. Subsequently, the cells were blocked with 2% BSA in PBS. The cells were then incubated with antibodies against flagellin and integrin-β2 and subsequently incubated with FITC-conjugated or Alexa 633-conjugated secondary antibodies. Cell nuclei were stained with 10 µM Hoechst33342 for 10 min. The images were obtained using either a laser scanning confocal microscope (Nikon, C1, Tokyo, Japan) or a fluorescent microscope (Olympus, CKX53, Tokyo, Japan) and analyzed using the ImageJ software version 1.52a (National Institutes of Health, Montgomery, MD, USA).

### 2.12. Nuclear Factor-Kappa B (NF-κB)/Activator Protein 1 (AP-1) Activation Reporter Assay

To measure NF-κB activation, THP-1 Xblue (InvivoGen, San Diego, CA, USA), a reporter cell expressing the embryonic alkaline phosphatase gene under the control of a promoter inducible by the transcription factor NF-κB and AP-1, was used. Cells were seeded onto 24-well culture plates at 0.8 × 10^6^ cells/well or 96-well culture plates at 2 × 10^5^ cells/well and stimulated with different treatment conditions. After treatment, a 20 µL aliquot of the supernatant was collected and added to 180 µL of a Quanti-Blue alkaline phosphatase detection medium for color development at 37 °C. After 2 h of color development, absorbance was measured using an ELISA reader (Bio-tek Instruments, Winooski, VT, USA) at 630 nm.

### 2.13. ELISA

Following specific treatments, the cell culture supernatants were collected, and the quantification of secreted IL-1β was performed using an IL-1β ELISA kit according to the manufacturer’s instructions (Biolegend, San Diego, CA, USA). Absorbance was measured using an ELISA reader at 450 nm. 

### 2.14. Statistical Analysis

The experiments were conducted at least three times, and all data are shown as the mean ± standard deviation (SD). Significant differences among groups were analyzed by one-way analysis of variance (ANOVA) followed by a post hoc test using the SPSS v.12.0 software. Differences were considered statistically significant at *p*-values of less than 0.05.

## 3. Results

### 3.1. ATRA Enhances the Cell Surface Hsp90 Protein Level

Hsp90, the major cytosolic molecular chaperone, plays a crucial role in the proper folding of client proteins, thereby maintaining their cellular functions. Several studies have recently discovered the presence of Hsp90 either in the extracellular space or on the cell surface of immune cells [3,21,22]. However, its function has not been well studied. In this study, to gain a new insight into the role of Hsp90 in monocyte-derived macrophages, we first examined the expression level of Hsp90 in monocytic THP-1 cells stimulated with ATRA. THP-1 cells were treated with either 1 µM ATRA or dimethyl sulfoxide (DMSO) as a control for 6 and 24 h, followed by the assessment of Hsp90 mRNA and protein levels, respectively. To distinguish between the intracellular and cell surface protein levels, a biotinylation procedure was used to specifically purify plasma membrane Hsp90 and its interacting partners. The isolated surface or intracellular proteins were analyzed by Western blotting. As shown in Figure 1, the mRNA (Figure 1a,b), intracellular protein (Figure 1c,e) and secreted protein levels (Figure 1f,g) of Hsp90 were almost unaffected by ATRA, while the cell surface level of Hsp90 was remarkably enhanced (Figure 1d,e). This enhancing effect increased with increasing ATRA concentrations, and 1 µM ATRA induced the maximum level of cell surface Hsp90, whereas the intracellular protein level was not affected, as revealed by the cell surface biotinylation assay (Figure 1e). These results indicate that ATRA specifically enhances surface Hsp90 expression in THP-1 cells.

### 3.2. ATRA Induces the Cell Surface Hsp90 Complex Formation in a Time- and Concentration-Dependent Manner

Since the cytosolic Hsp90 function depends on its interaction with other proteins within the complex, we determined whether ATRA-induced cell surface Hsp90 enhances the Hsp90 function by enhancing its complex formation. To detect possible complexes with Hsp90 protein on the cell surface, the membrane protein cross-linker bis (sulfo-succinimidyl) suberate (BS3) was used. Western blot analysis revealed that along with a 90 kDa protein band representing the monomeric form of Hsp90, there was a thick upper band of approximately 250 kDa (Figure 2a, lanes 3 and 4), which was almost ten-fold that in the control group (Figure 2b), suggesting that Hsp90 forms a complex with other cell surface protein (s). On the other hand, the intracellular cross-linking experiment with membrane-permeable ethylene glycol-bis succinimidyl succinate (EGS) showed that ATRA does not enhance the level of the intracellular Hsp90 complex (Figure 2a, lanes 1 and 2). The enhancing effect of ATRA on cell surface Hsp90 complex formation was shown to be concentration and time dependent (Figure 2c–f). The 12 h treatment with ATRA did not alter the Hsp90 complex/monomer ratio; however, the maximum level of the complex was observed after 24 h. As shown in Figure 2g–f, the complex formation was suppressed by not only geldanamycin (GA) (Figure 2g,h), which is an inhibitor of total cellular Hsp90, but also by biotin-geldanamycin (GeB) (Figure 2i,j), a specific inhibitor of cell surface Hsp90. These results indicate that ATRA selectively induces the cell surface Hsp90 complex.

### 3.3. Cell Surface Hsp90 Complex Formation Depends on RAR/RXR Pathway and Intracellular Calcium Level

ATRA exerts its biological functions by binding to the nuclear receptors, RAR and RXR, and the activation of both RAR and RXR is required for the full activation of the ATRA signaling pathway [23]. Thus, we examined whether these nuclear receptors are involved in the induction of the cell surface Hsp90 complex. THP-1 cells were treated with RAR/RXR agonist or with ATRA in the presence of RAR/RXR antagonist, and cell surface Hsp90 complex formation was assessed by Western blot analysis (Figure 3a,b). Treatment with RAR agonist or RXR agonist alone resulted in a 4–5-fold increase in the cell surface expression of the Hsp90 complex (lanes 3 and 4), but it was much lower than that of cells treated with ATRA (lane 2). In contrast, RAR or RXR antagonist alone could completely suppress this complex formation, suggesting that the activation of both RAR and RXR is required for Hsp90 complex formation.

Since several studies have reported the role of calcium ions in protein oligomerization [24], we investigated whether calcium ions are also involved in Hsp90 complex formation. First, we determined whether the intracellular calcium level was induced by treatment with ATRA by flow cytometry (Figure 3c; Appendix A) and fluorescence microscopy (Figure 3d). The results showed a gradual increase in the calcium level, reaching a maximum level at 12 h and decreasing to basal level at 24 h after ATRA treatment. To verify whether this ATRA-induced intracellular calcium level is associated with an increase in Hsp90 complex formation, either exogenous calcium or different types of calcium blockers were used. As shown in Figure 3e,f, exogenously administered calcium ions led to the formation of the Hsp90 complex at almost the same levels as those in ATRA-treated cells (lane 4). By contrast, calcium channel blockers and calcium chelators significantly prevented Hsp90 from forming complexes (lane 3, 5, respectively). Altogether, our data demonstrated that ATRA enhances the Hsp90 complex formation in a calcium-ion-dependent manner through the RAR/RXR signaling pathway. 

### 3.4. Identification of the Hsp90 Complex Components

To identify the protein partners of cell surface Hsp90, the proteins extracted from ATRA-stimulated THP-1 cells were subjected to Co-IP using an anti-Hsp90 antibody, and the purified proteins were then separated by SDS-PAGE followed by Coomassie Blue staining (Figure 4a). Five main bands (indicated with arrows—Figure 4a, lane 1) were separately excised and identified by LC-MS/MS and matched to the known sequences. The identified proteins were then assigned to bioinformatic analysis to assess the contribution of the interactome of Hsp90 to cellular compartments and biological functions. The identification of Hsp70, apoprotein B and 40S ribosomal protein as Hsp90′s known interacting partners confirmed that our procedure is sufficient for the purification of Hsp90 interactors. A total of 104 unique proteins were identified as interacting partners of Hsp90 (Appendix A). Among them, 13% corresponded to plasma membrane proteins, indicating that Hsp90 may associate with different proteins to form complexes at the plasma membrane of human macrophages (Figure 4b). Not surprisingly, Hsp70 was assessed with the highest confidence (as represented by a high score), which is known to be the main co-chaperone molecule of Hsp90. Interestingly, integrin αMβ2 (also known as CD11b/CD18) was identified with the highest coverage among the newly found membrane Hsp90 interacting proteins (Appendix A). Noticeably, integrin β2 is known to be markedly expressed on the surface of macrophages and plays a crucial role in the immune response against microbial infection [25]. Therefore, αMβ2 integrin was selected as the most potential candidate partner of cell surface Hsp90 in our experiment. 

### 3.5. Validation of the Interaction between the Hsp90 Chaperone and Integrin αMβ2

The identified Hsp70 and αMβ2 integrins were then validated for their interaction with Hsp90 by immunoblotting of proteins co-purified with anti-Hsp90-, anti-Hsp70-, anti-αM integrin or anti-β2 integrin monoclonal antibodies. Both Hsp70 and αMβ2 integrins were co-purified with Hsp90, indicating the interaction between Hsp90 and these proteins (Figure 5). However, both αM and β2 integrins were not co-purified with Hsp70 (Figure 5b–d), suggesting that Hsp70 and αMβ2 integrin do not directly interact or interact weakly with each other but rather link to one another through Hsp90. These results indicate that the main components of the Hsp90 complex are Hsp90 and Hsp70 molecules. The interaction between cell surface Hsp90 and integrin αM/β2 was further confirmed by immunoprecipitating the biotinylated protein with anti-Hsp90 antibody, indicating the induced level of cell surface Hsp90-bound αM/β2 integrin in the presence of ATRA (Figure 5e).

Since Sriram et al. reported the binding capacity of Hsp70 to two calcium ions within the ATPase domain, which is thought to be associated with Hsp70 function [26], and we found that Hsp90 complex formation depends on the presence of calcium ions (Figure 3e,f), we determined whether Hsp90-coupled Hsp70 is bound by calcium ions. The calcium-binding assay results revealed that calcium ions bind to Hsp70 but not to Hsp90 (Figure 5f; Appendix A).

### 3.6. ATRA Triggers Bacterial Flagellin Internalization in an Hsp90-Dependent Manner

To test whether ATRA-stimulated THP-1 cells were capable of taking purified flagellins, the cells were treated with 1 µM ATRA for 24 h before being challenged with 100 ng/mL flagellin from S. Typhimurium for different time periods. The flow cytometry results showed that in the absence of ATRA, the level of internalized flagellin did not change over time. In contrast, the uptake of flagellin by THP-1 cells treated with 1 µM ATRA started to increase 12 h after flagellin challenge and reached a maximum level at 48 h after the challenge (Figure 6a; Appendix A). The uptake of flagellin was gradually enhanced with the increase in ATRA concentrations and reached a maximum level at 1 µM ATRA (Figure 6b; Appendix A). Noticeably, by employing a specific inhibitor of cell surface Hsp90 (GeB), we found that GeB almost completely blocked the uptake of flagellin by ATRA-stimulated THP-1 cells (Figure 6c; Appendix A). In accordance with the results obtained by flow cytometry, the confocal microscopy results also showed a marked increase in the uptake of flagellin by ATRA, which was completely suppressed by GeB (Figure 6d). These data indicate that cell surface Hsp90 plays an important role in flagellin internalization by ATRA-stimulated THP-1 cells.

We next sought the mechanism driving the flagellin uptake under in vitro conditions. The entry of cell surface receptors into the cytosol is generally mediated by three major pathways: clathrin- (dynamin), lipid raft- and caveolin-mediated pathways. To verify the underlying mechanism by which flagellin is internalized, we examined the intracellular flagellin levels in the presence of different specific inhibitors of the three endocytosis pathways by flow cytometry. Our data revealed that dynasore, a specific inhibitor of clathrin-mediated endocytosis, suppressed the uptake of flagellin, while the other inhibitors of caveolin and lipid raft-mediated endocytosis did not influence the level of intracellular flagellin (Figure 6e; Appendix A). To confirm the results of flagellin uptake, we performed experiments using both permeabilized and non-permeabilized cells. The results showed that at 12 h after treatment, the flagellin signals were detected on the cell surface, while little or no intracellular flagellin was detected. At 48 h, almost no flagellin signals were detected on the cell surface, while higher levels of intracellular flagellin were detected (Figure 6f, upper panels). To further confirm the results of Figure 6e, we examined flagellin signal after treatment with different endocytosis inhibitors (Figure 6f, middle and lower panels). Our results showed that the lipid raft and caveolin inhibitors did not cause significant reduction in intracellular flagellin, while the dynamin inhibitor (Dyngo^®^ 4a) significantly inhibited the flagellin uptake. These results indicate that flagellin is internalized into the cell mainly through the clathrin-mediated endocytosis pathway.

### 3.7. αMβ2 Integrin Acts as a Carrier for Bacterial Flagellin Internalization

Along with the identification of integrin αMβ2 as a new interacting partner of cell surface Hsp90, the observation that cell surface Hsp90 is associated with the uptake of flagellin brings us to the hypothesis that integrin αMβ2 is also involved in the internalization of flagellin. To clarify this speculation, we first examined the expression of integrin αMβ2 by the treatment of ATRA and flagellin. As depicted in Figure 7a–d (Appendix A), the induction of cell surface integrin αMβ2 by ATRA was further enhanced by flagellin treatment, indicating a close correlation between flagellin recognition and integrin activation. A 48 h treatment with flagellin led to a low level of cell surface Hsp90-bound integrin αMβ2, which was reversed in the presence of dynamin inhibitor Dyngo^®^ 4a, indicating the internalization of integrin αMβ2 through a dynamin-dependent way in the presence of flagellin. Furthermore, by blocking αM integrin using the blocking monoclonal antibody CD11b/Mac-1 (ICRF44), we found a suppressed level of intracellular flagellin (Figure 7e; Appendix A), supporting the possible role of αMβ2 integrin as a carrier of flagellin. To evaluate this possibility, a colocalization assay was used to visualize flagellin binding to integrin αMβ2. Upon incubation of ATRA-stimulated THP-1 cells with flagellin, the colocalization of flagellin with integrin β2 was clearly observed both on the cell surface and inside the cell, as shown by the merged yellow color in fluorescent microscope data (Figure 8b). On the other hand, Hsp90 and flagellin were colocalized only on the surface of THP-1 cells but not in the intracellular compartment (Figure 8a), implying that Hsp90 is not a direct carrier for the delivery of flagellin into the cytosol but only acts at the early stage to transfer flagellin to integrin αMβ2 molecules.

### 3.8. Integrin-Carried Flagellin Is Capable of Activating the Inflammasomes

Cytosolic flagellin has been shown to initiate an inflammasome pathway, leading to the activation of caspase 1, which cleaves pro-IL-1β for the subsequent secretion of mature IL-1β. Here, we confirmed the capacity of flagellin internalized by the Hsp90–integrin complex to activate the inflammasome. Our previous study revealed that the inhibition of Hsp90 suppresses TLR5 surface expression and subsequent activation of NF-κB [27]. Therefore, here, we questioned whether the observed effect in this study with the association of Hsp90 is seemingly related to TLR5-mediated NF-κB signaling. To this end, we treated THP1-XBlue cells with ATRA in the presence of the Hsp90 membrane-permeable inhibitors GA, radicicol (Rad) and tanespimycin (17-AAG) or the impermeable inhibitor GeB, followed by a challenge with flagellin. NF-κB activity was measured. Our results showed that the inhibitor of cytosolic Hsp90 significantly suppressed flagellin-enhanced NF-κB activity, while the Hsp90 impermeable inhibitor (GeB) did not alter NF-κB activity (Figure 9a). These data imply that cell surface Hsp90 is not associated with TLR5-mediated NF-κB signaling through the recognition of flagellin. 

Next, we examined the expression level and activation of caspase-1 following flagellin challenge. Our data showed remarkable upregulation of the caspase-1 mRNA gene after 12 h of challenge with flagellin in the presence of ATRA (Figure 9b). Consistent with the mRNA expression level, the active form of caspase 1 (p20) was elevated by flagellin treatment in ATRA-stimulated THP-1 cells, and this increase was suppressed by pre-treatment with GeB (Figure 9c,d).

Finally, the secretion of mature IL-1β in response to integrin αMβ2-mediated cytosolic flagellin was examined. The ATRA treatment upregulated IL-1β by approximately 3-fold compared to the control group, while treatment with 100 ng/mL flagellin for 24 h dramatically upregulated the IL-1β mRNA expression by almost 10-fold that in the control group (Figure 9e). Consistently, the secreted level of IL-1β was found to be highly enhanced by flagellin treatment in the presence of ATRA, and this enhancing effect was almost attenuated by Hsp90 inhibitors (Figure 9f). Taken together, these results indicate that cytosolic flagellin carried by integrin αMβ2 is sufficient to activate the inflammasomes signaling cascade, leading to the activation of caspase-1 and secretion of IL-1β.

In summary, we identified, for the first time, a new interacting partner of the cell surface Hsp90 in THP-1 macrophages: integrin αMβ2. This protein acts as a carrier to deliver the purified flagellin into the cytosolic compartment of THP-1 cells to activate the inflammasome that drives the caspase-1 cleavage and IL-1β release (Figure 10).

## 4. Discussion

The present study sheds light on the pivotal role of the cell surface Hsp90 complex in the active delivery of exogenous bacterial flagellins into the cytosol, leading to inflammasome activation in human macrophages. In addition to its main location of intracellular compartments, Hsp90 is also detected on the surface of various cell types [28] and in the extracellular space [29]. This extracellular Hsp90 functions in different cellular processes, including antigen presentation [21], cell motility and wound healing [29]. The active delivery of exogenous protein antigens to the cytosol in association with extracellular Hsp90 has been studied, where the exogenously administered Hsp90–peptide antigen complex could be taken up by APC for presentation to T cells, either through the classical MHC class II pathway [4] or the MHC class I pathway [2,22,30]. In an earlier study, cationic liposomes were used and seemed to be necessary for the introduction of Hsp90–peptide antigen complexes into the cytosol of the APCs [30]. However, later studies revealed the successful delivery of exogenously administered Hsp90–peptide antigen complex [2,22] or antigen alone [3] to the cytosol without such liposomes, indicating the existence of an active endogenous delivery system. Imai et al. used antigen alone instead of the exogenous Hsp90–peptide antigen complex and found that the translocation of extracellular antigen into the cytosol is reduced in Hsp90α-null DCs and in DCs treated with a specific Hsp90 inhibitor [3]. All these studies support our findings that extracellular Hsp90 plays a role in the uptake of exogenous antigens by APC. However, compared with these previous studies where they used non-microbial antigens, such as tumor antigens or ovalbumins, we reported here, for the first time, the uptake of bacterial protein antigens by macrophages in association with extracellular Hsp90.

Hsp90 exerts diverse functions by forming a complex with other proteins. The Hsp90 complex contains several co-chaperones, such as Hsp70, p23 and cdc37, and accessory proteins that regulate Hsp90 activity and client protein recognition [31]. The results obtained by Co-IP and LC-MS/MS analysis in our study indicate that the Hsp90 complex consists of the Hsp90 homodimer and Hsp70, and Hsp90 interacts with both Hsp70 and αMβ2 integrin. The Hsp90/Hsp70 machinery has been well characterized, in which Hsp90 indirectly interacts with Hsp70 through the Hsp70-Hsp90 organizing (HOP) protein, and Hop is thought to be essential as a co-chaperone to facilitate the transfer of client proteins between Hsp70 and Hsp90 [32]. However, in our study, HOP was not identified among the interacting partners of Hsp90, and Hsp70 was co-precipitated with Hsp90, indicating that Hsp70 could directly interact with Hsp90 to form a fully functional chaperone, as eukaryotic organelles, which have Hsp70 and Hsp90 but lack a Hop, and Hsp70 and Hsp90 physically interact directly [33]. Since Hsp70 typically recognizes the client proteins through short hydrophobic sequences rich in leucine residues [34], Hsp70, instead of HOP, may interact with the hydrophobic domain of bacterial flagellin, which is known to be recognized by its corresponding receptor TLR5 [35]. Taken together with the fact that Hsp90 dimerization is the initial step necessary for the function of the Hsp90 complex [36], we propose a model for the interaction and assembly of Hsp90 complex components as follows: ATRA induces cell surface Hsp90 dimer to form a complex with Hsp70, and integrin αMβ2 interacts with the Hsp90 complex for flagellin transport (Figure 10, upper dotted box). 

The involvement of αMβ2 integrin in the delivery of exogenous bacterial flagellin to the cytosol of human macrophages is one of the most noticeable findings of the present study. αMβ2 integrin, the major β2 integrin, is highly expressed on the surface of innate immune cells, such as macrophages. It plays a crucial role in various cellular functions, including adhesion, migration, phagocytosis and cytokine production [37]. In line with a previous report [38], ATRA enhanced the cell surface expression of β2 integrin in THP-1 cells in our study. The central role of β2 integrins in immunity is highlighted by the fact that patients who lack β2 integrin expression are particularly prone to bacterial infections [39], and ATRA enhances integrin activation in human macrophages [40]. In this regard, our finding that ATRA-induced αMβ2 integrin is involved in enhancing the inflammatory response to bacterial flagellin by acting as a transmembrane transporter may explain the reason for this.

In the present study, ATRA enhanced the intracellular Ca^2+^ levels, and exogenously administered Ca^2+^ enhanced the cell surface expression of the Hsp90 complex in THP-1 cells, while treatment with a Ca^2+^ channel blocker inhibited the expression of the Hsp90 complex. Calcium ions play a role in Hsp70 structure stabilization and activation [26], and enhanced cellular Ca^2+^ levels are crucial for Hsp90–Hsp70 complex formation [41]. Taken together, we suggest that ATRA-enhanced intracellular Ca^2+^ levels are fundamental to form a cell surface Hsp90–Hsp70 complex.

The activation of inflammasomes caused by cytosolic flagellin has been reported in the last decade. However, in these established systems, flagellin is delivered by the bacterial transport system, such as the T3SS of S. Typhimurium [42]. The purified free flagellin must be encapsulated with lipid vesicles and passively enter the cytosol by transfection [43]. To date, there have been no reports on the active delivery of flagellin into the cytosolic compartment of host cells by the host delivery system. Here, we demonstrated a novel mechanism to explain the activation of inflammasomes and the subsequent release of IL-1β in response to exogenous flagellin without any bacterial delivery systems or experimental delivery systems. Since inflammasomes are cytosolic multiprotein oligomers that recognize cytosolic ligands, flagellin delivered by the αMβ2 integrin must be released into the cytosol to activate the inflammasome. Although the mechanism by which endosomal flagellin is released into the cytosol is unknown, cytosolic Hsp90 may play a role in this process [44]. 

In the present study, ATRA did not affect the cell surface expression of monomeric Hsp90, but it enhanced both the cell surface expression of the Hsp90 complex and the uptake of flagellin by THP-1 cells. A subsequent study revealed that ATRA-induced uptake of flagellin was inhibited by specific inhibitors of cell surface Hsp90 complex components. These observations indicate that a certain delivery system is necessary for the uptake of flagellin by THP-1 cells in the absence of ATRA and that ATRA-induced cell surface Hsp90 complex is a prerequisite for the effective uptake of flagellin by THP-1 cells. One of the immunoregulatory functions of ATRA is to induce the differentiation of monocytes into macrophages [45]. During the differentiation of monocytes to macrophages, the levels of Hsp90 and Hsp70 increase [46]. The elevation of these Hsps has been suggested to play a role in the mechanism of differentiation from a monocyte to a macrophage phenotype [46]. In this regard, we suggest that the ATRA-induced cell surface Hsp90 complex and its delivery function are exclusively macrophage specific but not monocyte specific. However, our current study is limited to the monocytic THP-1 cell line; future studies on activated macrophage cell lines will be needed to confirm our finding.

The present study shows how ATRA enhances the immune response of human macrophages to bacterial flagellin. The importance of ATRA in maintaining resistance to infection can be recognized by an increase in susceptibility to infection in vitamin-A-deficient individuals. During the last two decades, the role of ATRA in the maintenance of immune homeostasis has been uncovered, providing insight into its importance for immunity, especially for resistance to infection. However, the precise molecular mechanism by which ATRA facilitates the recognition of PAMPs by host immune receptors has not yet been reported. Therefore, our results provide important insights into the mechanism by which vitamin A enhances the functional resistance of human macrophages to infectious agents by providing a novel mechanism for Hsp90 complex-facilitated and αMβ2 integrin-mediated bacterial flagellin delivery to the cytosolic pathogen recognition receptor, in addition to the surface of the cell.

## Figures and Tables

**Figure 1 cells-11-02878-f001:**
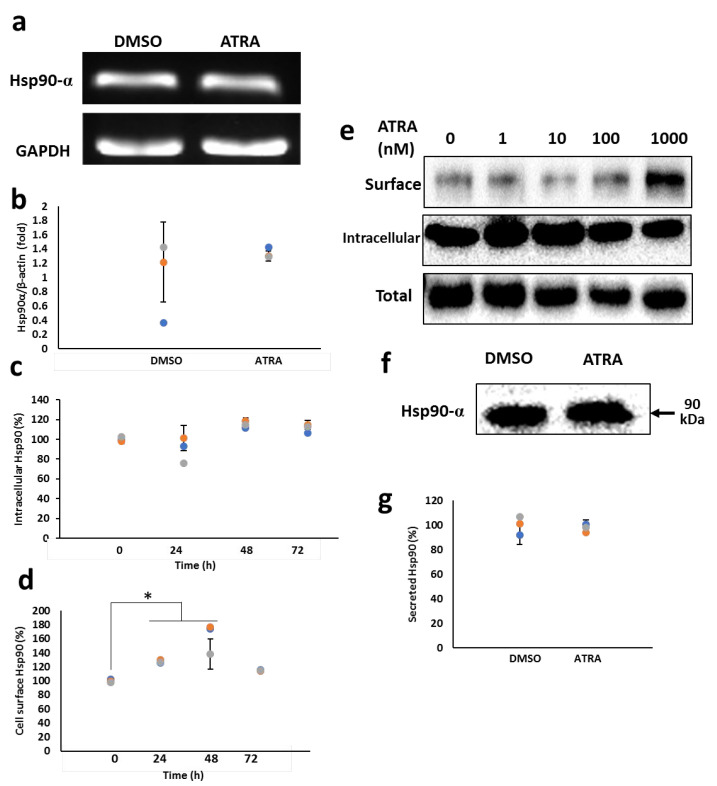
ATRA enhances the cell surface expression of Hsp90. (**a**,**b**) Hsp90 mRNA expression. THP-1 cells were treated with 1 µM ATRA for 6 h. Hsp90 mRNA level was examined by semi-qPCR (**a**) and real-time qPCR (**b**). (**c**,**d**) Protein expression of Hsp90. Cells were treated with 1 µM ATRA for various time points. The intracellular (**c**) or cell surface (**d**) levels of Hsp90 were measured by flow cytometry (Appendix A). (**e**) Biotinylation and Western blotting of Hsp90. Cells were treated with various concentrations of ATRA as indicated for 24 h. Then, the cells were treated with sulfo-succinimide-biotin. Bound proteins were then eluted and immunoblotted with the anti-Hsp90 antibody. (**f**,**g**) Secreted Hsp90 levels. Cells were treated with 1 µM ATRA for 48 h. The levels of secreted Hsp90 were measured by Western blotting (**f**) or ELISA (**g**). * *p* < 0.05 vs. control group.

**Figure 2 cells-11-02878-f002:**
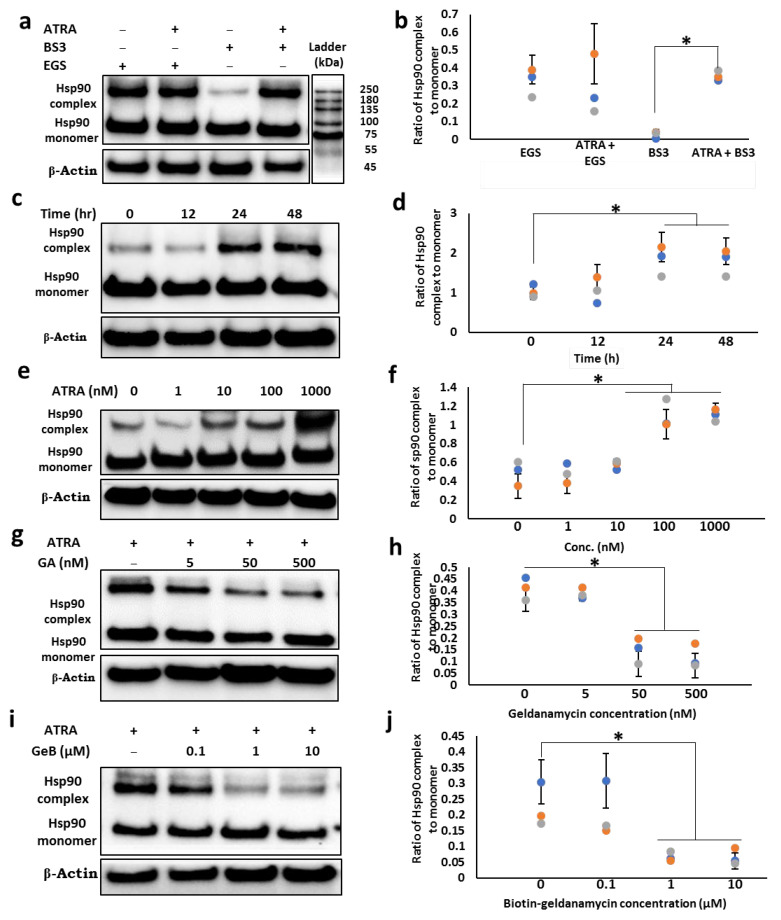
ATRA enhances the cell surface Hsp90α complex formation. (**a**,**b**) Detection of Hsp90 complex using either cell impermeable cross-linker BS3 or permeable cross-linker EGS. (**c**–**j**) Cell surface Hsp90 complex in different experimental conditions. THP-1 cells were treated with 1 µM ATRA for 24 h (**a**,**b**) or with 1 µM ATRA for different time periods (**c**,**d**), with different concentrations of ATRA for 24 h (**e**,**f**) or in the presence of Hsp90 permeable inhibitor (**g**,**h**) or Hsp90 impermeable inhibitor (**i**,**j**). The cells were then incubated with either 2.5 mM EGS (**a**,**b**) or 2.5 mM BS3 (**a**–**j**) for 30 min. After the cross-linking reaction, the collected lysate was analyzed by immunoblotting with an antibody against Hsp90. (**b**,**d**,**f**,**h**,**j**) Densitometric analysis of the protein bands was performed using the ChemiDoc MP Imaging System (n = 3). The ratios of Hsp90 oligomers to monomers were calculated, and data were normalized to the control condition (DMSO). The ratio of oligomer/monomer in the control group was set to 100%. Statistical analyses were performed between the control and treated groups; * *p* < 0.05.

**Figure 3 cells-11-02878-f003:**
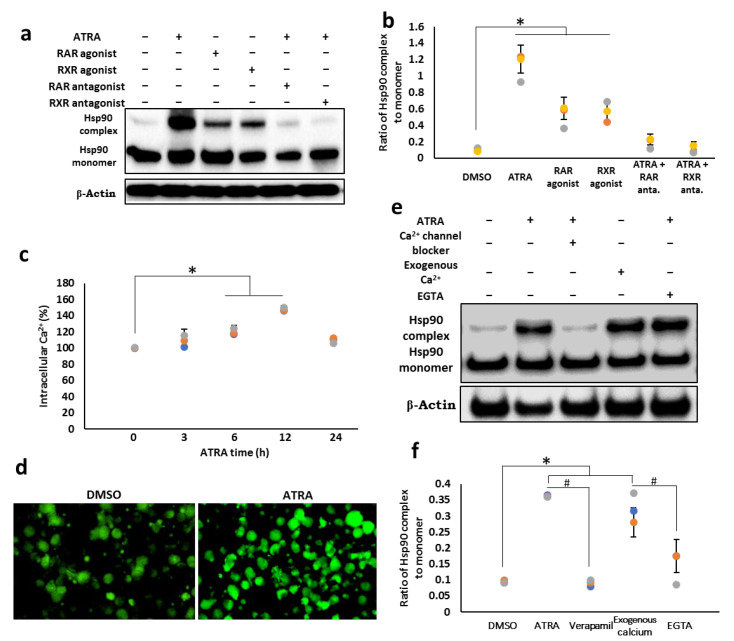
Cell surface Hsp90 complex formation depends on RAR/RXR pathway and intracellular calcium level. (**a**,**b**) Cell surface Hsp90 complex in the presence or absence of RAR/RXR agonists or antagonists. THP-1 cells were treated with 1 µM RAR or RXR antagonists in the presence of 1 µM ATRA, or with 1 µM RAR or RXR agonists without ATRA for 24 h. Cells were then collected and subjected to cross-linking reaction with BS3. The cell lysates were analyzed by Western blotting. (**c**,**d**) Intracellular calcium levels. Cells were treated with 1 µM ATRA for various time points as indicated, and the intracellular calcium level was measured by flow cytometry (**c**), or the cells were treated with 1 µM ATRA for 12 h, and the intracellular calcium level was determined by fluorescence microscope (**d**). (**e**,**f**) Cell surface Hsp90 complex in the presence or absence of calcium channel blocker or the chelator. Cells were treated with either the calcium channel blocker, verapamil or the chelator, ethyleneglycoltetraacetic acid (EGTA), in the presence of 1 µM ATRA or with 2.5 mM exogenous calcium without ATRA. Cells were then collected and subjected to the cross-linking reaction. The cell lysates were analyzed by Western blotting. (**b**,**f**) Densitometric analysis of protein bands was performed with the ChemiDoc MP Imaging System (n = 4). The ratios of Hsp90 oligomer/monomer were calculated, and the data were normalized to the DMSO control. The ratio of oligomer/monomer from the control group was set as 100%. Statistical analysis was performed between control and treated groups (n = 3). * *p* < 0.05; # *p* < 0.05.

**Figure 4 cells-11-02878-f004:**
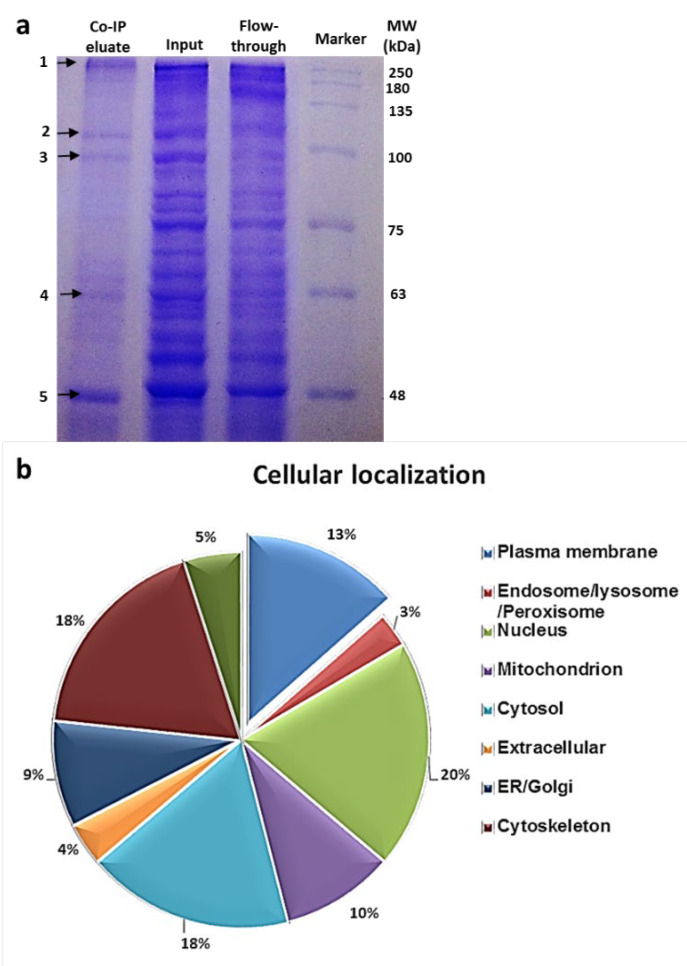
Identification of the Hsp90 interacting partners. (**a**) SDS-PAGE and Coomassie blue staining of proteins obtained in co-immunoprecipitation assay with anti-Hsp90α antibodies using cell lysates prepared from ATRA-stimulated THP-1 cells. Immunoprecipitated proteins (lane 1) or total cell lysates (lane 2) or unbounded proteins (lane 3) were separated by SDS-PAGE and visualized by Coomassie blue staining. Five main bands (indicated by arrows) obtained via Co-IP were eluted separately and analyzed by LC-MS/MS. (**b**) Cellular localization of Hsp90 interacting proteins by bioinformatic analysis. Hsp90 interacting partners identified by LC-MS/MS were analyzed for cellular localization. The identified proteins were classified based on their subcellular localization, of which 13% were plasma membrane proteins.

**Figure 5 cells-11-02878-f005:**
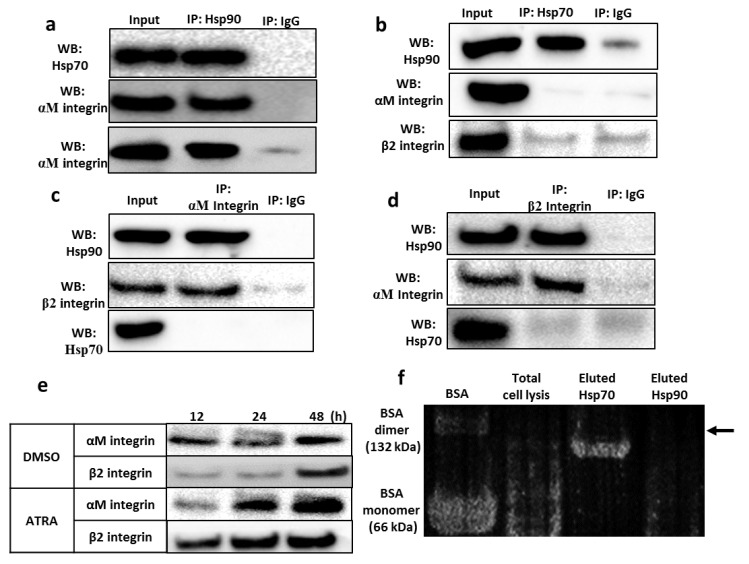
Hsp90 interacts with Hsp70 and integrin αMβ2 in the cell surface complex of THP-1 cells. (**a**–**d**) Coimmunoprecipitation (Co-IP) was performed from whole-cell lysate of ATRA-stimulated THP-1 cells with specific antibodies against Hsp90 (**a**), Hsp70 (**b**), αM integrin (**c**) or β2 integrin (**d**). The precipitates were immunoblotted with antibodies indicated on the left of panels. IgG represents a control antibody used for IPs. (**e**) Cell surface Hsp90-bound integrin αMβ2 was induced by ATRA treatment. THP-1 cells were treated with 1 µM ATRA for different time points. The cell surface protein was biotinylated, immunoprecipitated with anti-Hsp90 antibody and immunoblotted with anti-αM/β2 integrin antibodies. (**f**) Calcium-binding capacity of Hsp70. The band illustrating Hsp70 (lane 3) or Hsp90 (lane 4) or the total cell lysis (lane 2) was eluted from the gel and run onto SDS-PAGE followed by membrane transfer. BSA was used as a positive control (lane 1). Membrane was then incubated with the developing solution for detecting calcium-binding protein.

**Figure 6 cells-11-02878-f006:**
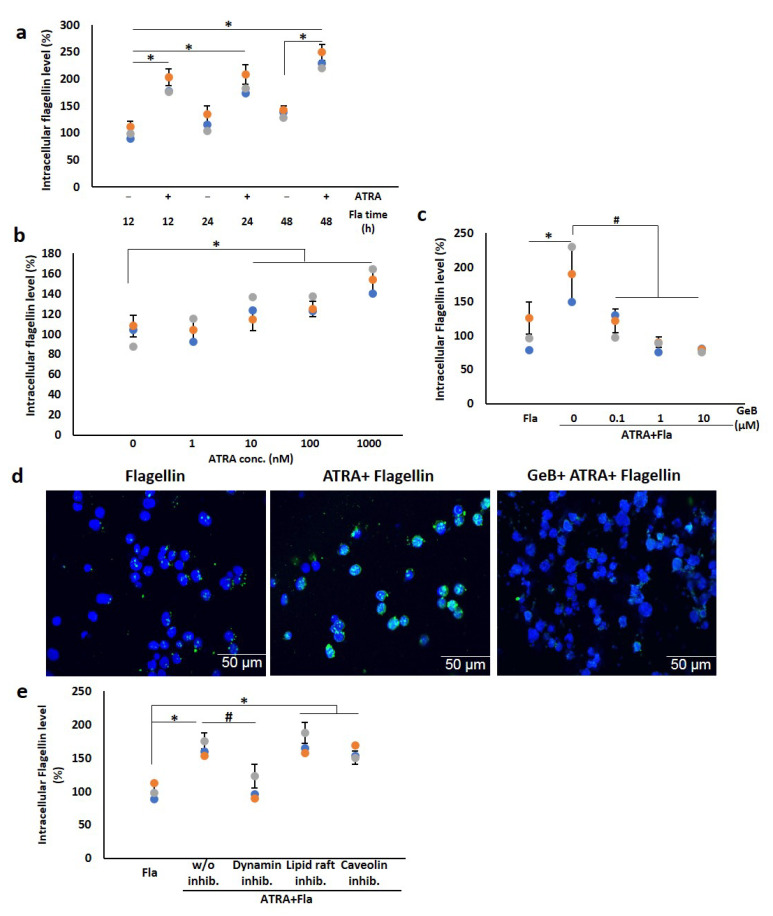
ATRA enhances the Hsp90-mediated flagellin internalization via the clathrin-mediated endocytosis pathway. (**a**,**b**) Intracellular flagellin levels following treatment with ATRA and flagellin at different exposure times and concentrations of ATRA. THP-1 cells were treated with 1 µM ATRA for 24 h or with DMSO followed by challenge with 100 ng/mL flagellin for different time periods or with various concentrations of ATRA for 24 h before 48 h incubation with 100 ng/mL flagellin. (**c**,**d**) Intracellular flagellin levels in the presence of specific inhibitor of cell surface Hsp90 (GeB). Cells were exposed to different concentrations of GeB for 2 h prior to stimulation with 1 µM ATRA for 24 h and 100 ng/mL flagellin for 48 h. The intracellular level of flagellin was measured by flow cytometry (**c**) or by confocal microscopy (**d**). After 24 h stimulation with 1 µM ATRA in the presence (right panel) or absence (left and middle panel) of the Hsp90 inhibitor (GeB), the cells were further treated with 100 ng/mL flagellin for 48 h. The collected cells were then fixed with formaldehyde, permeabilized with Triton X-100 and incubated with the anti-flagellin antibody overnight at 4 °C. After subsequent incubation with the fluorescein isothiocyanate (FITC)-conjugated secondary antibody and Hoechst33342 staining dye for nucleus, the intracellular level of flagellin was detected by confocal microscope (×200). Scale bars represent 50 µm. (**e**) Intracellular flagellin levels in the presence of different endocytosis inhibitors. Cells were stimulated with 1 µM ATRA for 24 h prior to treatment with different endocytosis inhibitors (10 µM dynasore (dynamin inhibitor), 5 mM methyl-β cyclodextrin (lipid raft inhibitor) or 1.5 µM filipin III (caveolin inhibitor)) for 2 h. The cells were further treated with 100 ng/mL flagellin for 48 h. The flagellin levels were measured by flow cytometry. (**f**) Intracellular and cell surface flagellin levels in the presence of different endocytosis inhibitors. Cells were treated with 1 µM ATRA in the presence or absence of different endocytosis inhibitors (1 µM Dyngo^®^ 4a (dynamin inhibitor), 5 mM methyl-β cyclodextrin (lipid raft inhibitor) or 1.5 µM filipin III (caveolin inhibitor)) for 2 h, followed by the treatment with 100 ng/mL flagellin for 1 h or 48 h. The intracellular (right panel) or cell surface flagellin levels (left panel) were measured by fluorescence microscopy (×200). Green, flagellin; blue, nuclei. * *p* < 0.05; # *p* < 0.05.

**Figure 7 cells-11-02878-f007:**
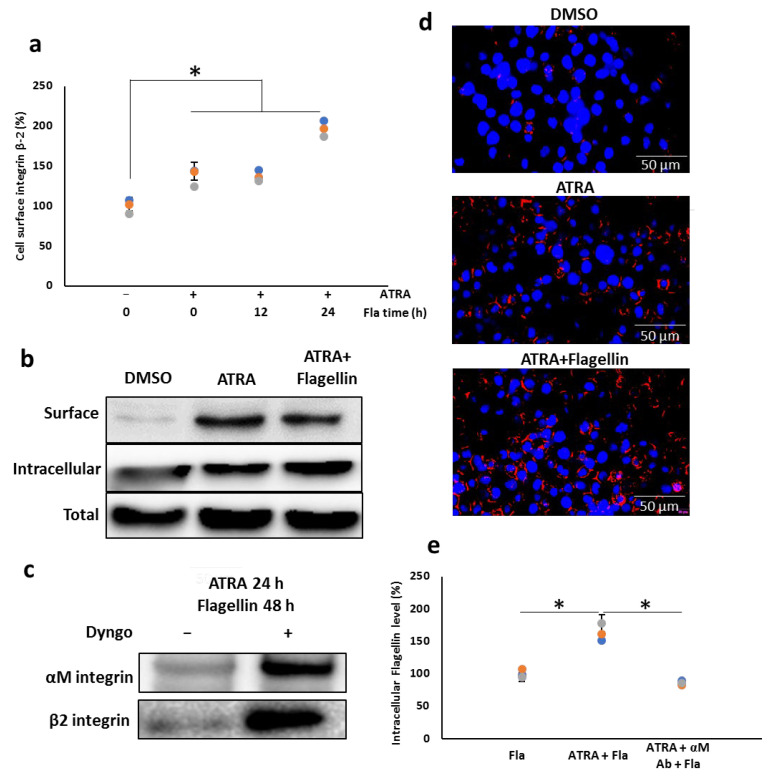
Flagellin enhances the cell surface integrin αMβ2 expression, and the treatment with the anti-αM integrin antibody reverses the ATRA-enhanced flagellin uptake. (**a**) Cell surface levels of integrin β2 examined by flow cytometry. THP-1 cells were stimulated with 1 µM ATRA for 24 h followed by treatment with 100 ng/mL flagellin for different time periods, and cell surface integrin β2 was examined by flow cytometry. (**b**) Integrin β2 expression examined by Western blot analysis. Cell surface integrin β2 was purified by biotinylation, and the surface proteins, intracellular proteins or total proteins were subjected to Western blotting. (**c**) Cell surface Hsp90-bound αM/β2 examined by Western blot. THP-1 cells were treated with 1 µM ATRA for 24 h in the presence or absence of 1 µM dynamin-specific inhibitor Dyngo^®^ followed by 48 h treatment with flagellin. The cell surface protein was biotinylated, immunoprecipitated with anti-Hsp90 antibody and immunoblotted with anti-αM/β2 integrin antibodies. (**d**) Intracellular integrin β2 expression examined by fluorescence microscope. To detect intracellular integrin β2, the collected cells were fixed with formaldehyde, permeabilized with Triton X-100 and incubated with the anti-integrin β2 antibody overnight at 4 °C. After subsequent incubation with the Alexa 633-conjugated secondary antibody and Hoechst33342 staining dye for nucleus, the level of integrin β2 was detected by fluorescence microscope (×200). Scale bars represent 50 µm. (**e**) Intracellular integrin β2 expression examined by flow cytometry. THP-1 cells were stimulated with 1 µM ATRA for 24 h, followed by 2 h treatment with 1 µg/mL anti-αM integrin antibody prior to treatment with 100 ng/mL flagellin for 24 h. The intracellular level of flagellin was measured by flow cytometry. * *p* < 0.05.

**Figure 8 cells-11-02878-f008:**
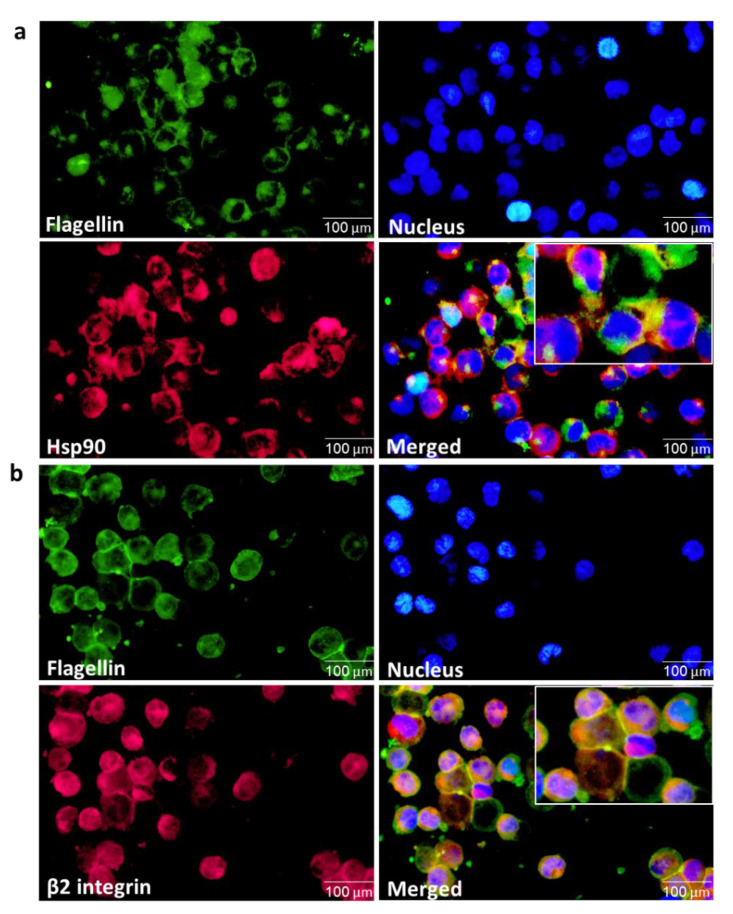
Flagellin colocalizes with β2 integrin but not with Hsp90 inside the cells. THP-1 cells were stimulated with 1 µM ATRA for 24 h, followed by incubation with 100 ng/mL flagellin for 48 h. The collected cells were then fixed with formaldehyde, permeabilized with Triton X-100 and incubated with the FITC-conjugated anti-flagellin and Alexa 633-conjugated anti-Hsp90 antibodies (**a**) or with the anti-flagellin and Alexa 633-conjugated anti-integrin β2 antibodies overnight at 4 °C (**b**). The cellular distribution of given proteins was detected by fluorescence microscope (×400). Scale bars represent 100 µm.

**Figure 9 cells-11-02878-f009:**
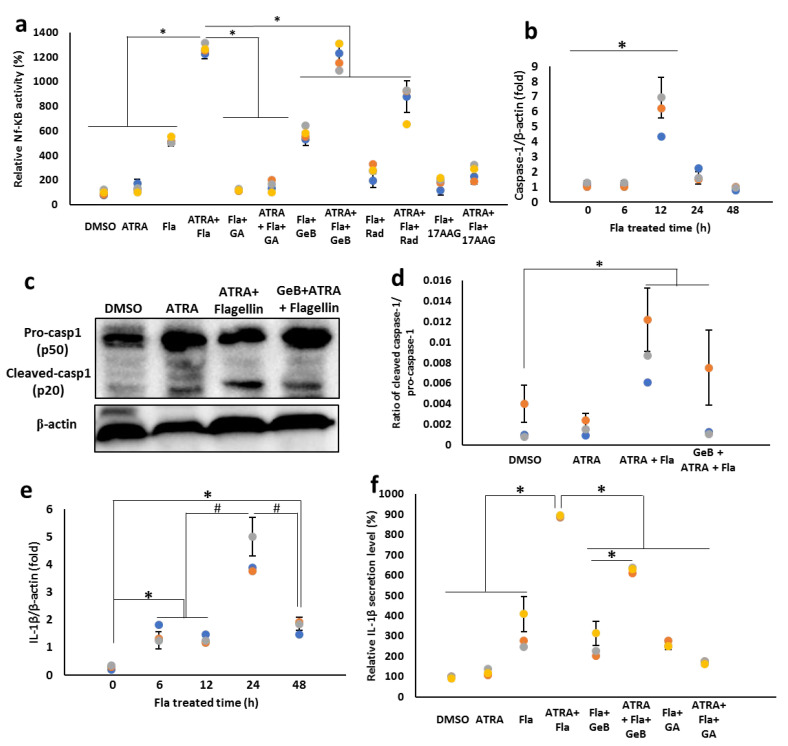
Exogenously administered flagellin activates caspase-1 and induces IL-1β secretion in an Hsp90-dependent manner. (**a**) NF-κB activity. THP-1 Xblue cells were stimulated with 1 µM ATRA in the presence or absence of different Hsp90 inhibitors, followed by 48 h treatment with 100 ng/mL flagellin. After stimulation, a 20 µL aliquot of the supernatant was collected and added to 180 µL of Quanti-Blue medium for color development at 37 °C. After 2 h color development, absorbance measurements were performed using an ELISA plate reader at 630 nm. Bar graphs indicate the optical density. (**b**,**c**) Caspase-1 mRNA and protein expression. To measure mRNA expression, THP-1 cells were treated with 1 µM ATRA for 24 h, followed by treatment with 100 ng/mL flagellin for different time periods (**b**,**e**). The mRNA levels of caspase-1 (**b**) and IL-1β (**e**) were measured by real-time qPCR. To determine protein expression, cells were stimulated with 1 µM ATRA for 24 h, followed by treatment with 100 ng/mL flagellin for 48 h. After stimulation, caspase-1 activation and IL-1β secretion were measured by immunoblotting (**c**) and ELISA (**f**), respectively. (**d**) The ratio of densitometry values of cleaved caspase-1 with pro-caspase-1. THP-1 cells were stimulated with 1 µM ATRA in the presence or absence of cell surface Hsp90 inhibitor GeB, followed by 48 h treatment with 100 ng/mL flagellin. Densitometric analysis of the protein bands was performed with the ChemiDoc MP Imaging System (n = 3). (**e**,**f**) IL-1β mRNA expression and secretion. * *p* < 0.05; # *p* < 0.05.

**Figure 10 cells-11-02878-f010:**
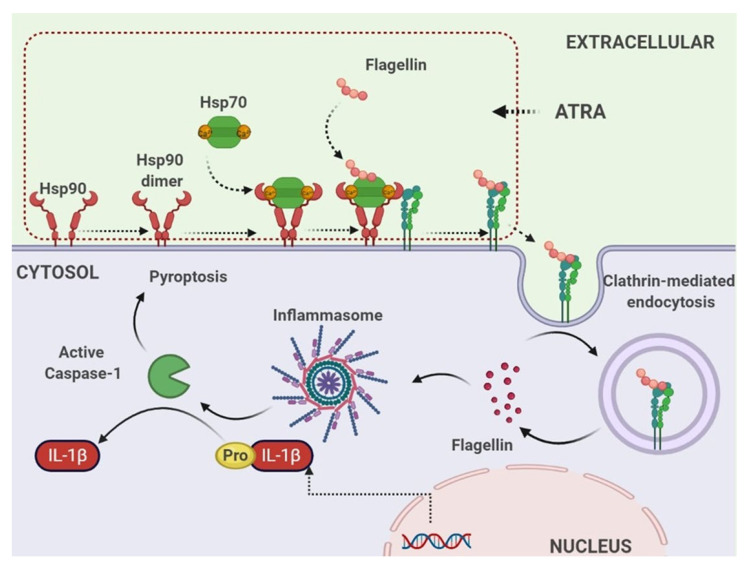
Proposed mechanism underlying the Hsp90/αMβ2 integrin complex-mediated flagellin internalization and intracellular recognition in human monocyte THP-1 cells treated with ATRA. The stimulation of ATRA in THP-1 cells results in the increase in the level of cell surface Hsp90. Upon the binding of Ca^2+^, Hsp70 binds the cell surface Hsp90 dimer. Functional Hsp90 chaperone, in turn, interacts with the surface αMβ2 integrin to form the Hsp90/αMβ2 integrin complex, which subsequently internalizes the bacterial flagellin for recognition by the inflammasome. As a result, these upstream events induce IL-1β secretion and cell death by pyroptosis.

## Data Availability

All data generated or analyzed during this study are included in this published article and its Appendix A files.

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
