# Peer review of "Cell Surface Hsp90- and αMβ2 Integrin-Mediated Uptake of Bacterial Flagellins to Activate Inflammasomes by Human Macrophages"

_cells, 2022, doi:10.3390/cells11182878_

Round 1

Reviewer 1 Report

In this study, Thi Xoan Hoang and Jae Young Kim provide new insights into a well known PAMP internalization by macrophages. The authors provide a detailed study on the recently identified surface Hsp90 complex and the signaling pathway leading to inflammation, as well as the role of ATRA on the cell surface Hsp90 abundance. Finally, the authors suggest that αMβ2 integrin mediate the flagellin uptake. Overall, the manuscript is well written, easy to read and well structured.

I have minor comments:

1-      In figure 5f, the control for Hsp90 (using the Hsp90 antibody) is missing. The arrow is not sufficient.

2-      L384: The authors conclude that Hsp90 interact directly with Hsp70 and αMβ2 in-383 integrins. However, the co-immunoprecipitation is done using a whole cell lysate which leaves the possibility of an intermediate partner. The authors should remove this sentence.

3-      The data have all ben obtained in one cell line, THP1. If THP-1 Is a reliable to study macrophages pathway, the authors should highlight in the discussion that these findings need to be confirmed in:

- activated macrophages (which is most likely what macrophages would experience as during S. enterica infections, there would be other PAMPs, including LPS)

- in other macrophages cell lines such as primary macrophages in which for example other host sensors (Naip) have been found important to sense cytosolic flagellin.

Author Response

Dear Reviewer,
Thank you for your kind and considerate comments, which have greatly helped improve the quality of our current version. We have carefully studied your comments and tried our best to address them one by one, and the revised parts have been highlighted in red for your kind consideration. We hope the revised manuscript meets your requirements.

Please see the attachment below that contains our point-to-point response to the reviewer.

Sincerely,

Thi Xoan Hoang

Reviewer 2 Report

Very well designed study and well written manuscript. 

"Hsp90-" in the title. I guess, author wanted to mention Hsp90a.

  Introduction was a bit too much with irrelevant text which has no connection with results presented.   THP-1 texts from line 97 and onwards are repititive. All these texts were mentioned in line 103.   L-134, "5" should be superscript   It was not mentioned in figure legends, how many times the experiment was repeated.   Flow cytometry data needs to be given as histogram as well.  

Bands in Fig 5F are not clear. 

Author Response

Dear Reviewer,

Thank you for your kind and considerate comments on our manuscript. The authors have carefully studied your comments and tried to address them one by one, which we hope to meet your requirement. Revised portions are highlighted in the revised manuscript for your kind consideration.

Please see the attachment below that contains our point-to-point response to the reviewer.

Sincerely,

Thi Xoan Hoang, PhD
